# Body Composition and Physical Fitness in Madeira Youth

**DOI:** 10.3390/children9121833

**Published:** 2022-11-27

**Authors:** Diogo V. Martinho, Élvio Rúbio Gouveia, Cíntia França, Helder Lopes, Andreas Ihle, Adilson Marques, Ana Rodrigues, Ricardo Alves, Hugo Sarmento

**Affiliations:** 1University of Coimbra, Research Unit for Sport and Physical Activity (CIDAF), Faculty of Sport Sciences and Physical Education, 3040-248 Coimbra, Portugal; 2Department of Physical Education and Sport, University of Madeira, 9020-105 Funchal, Portugal; 3Laboratory of Robotics and Engineering Systems (LARSYS), Interactive Technologies Institute, 9020-105 Funchal, Portugal; 4Center for the Interdisciplinary Study of Gerontology and Vulnerability, University of Geneva, 1205 Geneva, Switzerland; 5Research Center in Sports Sciences, Health Sciences, and Human Development (CIDESD), 5001-801 Vila Real, Portugal; 6Department of Psychology, University of Geneva, 1205 Geneva, Switzerland; 7Swiss National Centre of Competence in Research LIVES—Overcoming Vulnerability: Life Course Perspectives, 1015 Lausanne, Switzerland; 8CIPER, Faculty of Human Kinetics, University of Lisbon, 1495-751 Lisbon, Portugal; 9Environmental Health Institute (ISAMB), Faculty of Medicine, University of Lisbon, 1649-020 Lisbon, Portugal

**Keywords:** body fat, cardiovascular fitness, health promotion, physical activity

## Abstract

Background: Research on composition and physical fitness is lacking in school-aged children from Madeira. This study aimed to examine the frequencies of overweight and obese participants and, in addition, to compare the fitness levels of Madeira youth with reference data. Methods: The sample comprised 521 participants (n = 258 boys; n = 263 girls) aged 10.0–18.9 years old. Methods: Height and weight were measured. Body mass index (BMI) was calculated, and percent fat was derived from skinfolds. Tests detailed on FITESCOLA battery were performed. BMI was plotted against U.S. reference data and physical assessment contrasted with the cut-off points of FITESCOLA protocol and corresponding data developed in Portuguese youth. Results: The percentages of overweight and obesity were 15% (overweight) and 14% (obesity) for boys and 16% (overweight) and 6% (obesity) for girls. The median values of fat mass percentage were closely related to the cut-off standards; however, substantial variation was noted. Boys and girls tended to be below the recommendations of cardiorespiratory fitness. Conclusions: Sport participation should be promoted in Madeira to attain acceptable values of body composition and physical fitness levels.

## 1. Introduction

Over the last few decades, the prevalence of overweight and obesity increased significantly among youth and is often described as a worldwide epidemic [1,2]. According to a recent World Health Organization (WHO) report, the prevalence of overweight and obesity from 1975 to 2016 raised 5% and 7% in males and females, respectively [3]. In Portugal, data from National Food, Nutrition, and Physical Activity Survey (2015–2016) defined pre-obesity and obesity according to body mass index (BMI). Based on BMI, approximately 17% and 24% of children and adolescents were classified as pre-obesity, respectively. The prevalence of obesity was 7.7% for children and 8.7% for adolescents [4]. Although the survey showed the highest values of pre-obesity (35.9%) and obesity (22.3%) in the Madeira population [4], data about the prevalence of overweight and obesity in Madeira youth are limited to pre-pubertal children [5]. The prevalence of obesity was 27.0% Among 168 males and 155 females aged 9–10 years old from the Madeira region [6]. In addition, results from a cross-sectional survey of 1258 boys and 1126 girls aged 6–10 years old noted that 14.1% males and 17.1% females were classified as overweight based on United States Centers for Disease Control and Prevention criteria, whereas 17.3% males and 14.4% were obese [7]. More recently, a previous survey in 22048 Portuguese 10–18-year-old participants about the prevalence of overweight and obesity did not consider the Madeira adolescents [8]. Nevertheless, adiposity was associated with an increase in cardiovascular risk factors [9,10] and a decrease in time spent participating in moderate-to-vigorous physical activity (MVPA) [11,12].

Schools play a central role in assessing body composition and physical fitness levels as children and adolescents spend much of their time at school [13]. In Portugal, the batteries recommended by the National Physical Education Program that are widely applied in schools are the FITNESSGRAM, and more recently, the FITESCOLA [14,15]. Although different protocols were used in FITNESSGRAM and FITESCOLA, both batteries examine body composition, neuromuscular and cardiorespiratory fitness of youth participants [15,16] and, in addition, present age and sex-specific cut-off points. Appropriate levels of physical fitness in youth age ranges are associate with future health parameters [17] and, consequently, data of fitness indicators are commonly described in several countries [18,19,20,21]. For example, cardiorespiratory fitness tended to decrease from 8 to 14 years old in Canadian males, whereas among girls grip strength decreased from 11 to 19 years old [20]. During the last decade the levels of fitness in German youth remained stable [21]. Data among American youth are not consistent. The prevalence of the highest level of fitness increased 7.8% in boys and girls aged 9–19 and longitudinally followed 2006–2017 [18]. In contrast, a survey of 192,848 participants aged 9–19 years old reported a trivial percentage (i.e., <13%) of participants that attained healthy fitness in six protocols included on FITNESSGRAM battery [19]. In Portugal, secular trends of physical fitness tests (i.e., sit and reach, curl-ups, horizontal jump, velocity) indicated that from 1993 to 2013, boys and girls improved physical fitness [22]. Although 61% of participants who resided in mainland Portugal were classified as fit for the 20 m shuttle in a cross-sectional sample of 22,048 children [14], recent data estimated a considerable decline in CRF in recent years [23].

The interrelationship among body composition and fitness in Portuguese youth was documented in 86 Portuguese participants (44 females and 42 males) who participated in the European Youth Heart Study in 2000 and completed a follow-up evaluation ten years later [24]. Variation in cardiorespiratory fitness explained 39% and 26% of variance in changes of fat mass and waist circumference, respectively [24]. Among 2696 Portuguese adolescents, cardiovascular fitness and push-ups emerged as the main predictors of BMI and waist circumference [15]. Additionally, cardiorespiratory fitness mediated the negative association between time spent in MVPA and adiposity [25]. Given the association of fitness and adiposity during the first two decades of life with health in adulthood [26] it is crucial to describe these parameters, comparing them with the cut-off points proposed by batteries applied on the school. From a public health perspective, monitoring physical fitness and body composition is a priority, particularly in demographic regions where the studies are scarce. Of note, previous studies that contested body size and fitness with FITNESSGRAM battery involved participants from Portugal Midlands [15] and Azores island [27].

Therefore, the purposes of this study were to examine body composition and physical fitness in a sample of males and females from the Autonomous Region of Madeira and, in addition, to contrast the mentioned data with cut-off points defined by FITESCOLA and Portuguese-specific values.

## 2. Methods

### 2.1. Procedures and Approval

The present study was conducted in five public schools in the capital of Madeira Archipelago. According to the provisory census in 2021, Funchal comprises 105,794 residents [28]. Statistics from the national observatory obtained in 2020–2021 estimated that 4443 students were registered in the 1st cycle, 2627 in the primary school, 4465 in the 3rd cycle, and 7894 in the secondary schools [29]. The data were collected in 2016–2019 and is part of the project “Physical Education in School of Autonomous Region of Madeira—Understanding, Intervening, Transforming (EFERAM-CIT)”, which was approved by the University of Madeira (ACTA N.77-12/April/2016). This study was also supported by the Regional Secretary of Education. The study followed the recommendations of the Helsinki Declaration produced by the World Medical Association (2013) [30] for research with humans. Parents, legal guardians, and participants were informed about the aim, procedures, and risks of the present study. Participation was voluntary, and participants could withdraw from the study anytime. Measurements were obtained by teachers of Physical Education or Master’s degree students who were familiarized with the anthropometry and battery of fitness tests. Before data collection, teachers and students were trained in anthropometric procedures, fitness assessment, and data organization. 

### 2.2. Sample and Chronological Age

The initial database included 616 school participants recruited from the five public schools in the capital of the Madeira Archipelago. However, 95 participants did not complete the anthropometric measurements, fitness tests, or the questionnaire about participation in organized or school sports and, consequently, were excluded. Thus, the final sample comprised 521 participants (n = 258 boys; n = 263 girls) aged 10.0–18.9 years old. The age ranges of boys and girls were 10.0–18.7 and 10.0–18.9 years old, respectively. Chronological age was calculated as the difference between the day of assessments at the school and the birthday. Participants were grouped by age bands of 1.0 years.

### 2.3. Anthropometry

Participants were measured wearing shorts and shoes were removed. Height was obtained to the nearest 0.1 cm using a portable stadiometer (SECA 213, Hamburg, Germany). Weight was measured to the nearest 0.1 kg with a mechanical scale (SECA 762, Hamburg, Germany). Body mass index (BMI) was calculated, and participants were individually plotted relative to reference data for American children and adolescents [31]. Based on a percentile range, participants were classified as follows: underweight (<5th percentile), healthy weight (5th ≤ percentile < 85th), overweight (85th ≤ percentile < 95th), obesity (≥95th percentile). Waist circumference was measured just above the lateral border of the ilium to the nearest 0.1 cm using a steel measuring tape. Two skinfolds (i.e., triceps and calf) were measured to the nearest 0.1 mm with a caliper (Harpenden, UK). Sex-specific equations developed for youth [32] were adopted by the FITESCOLA battery and therefore used to determine fat mass percentage (%FM). The equation is frequently applied in samples of children and adolescents [15,27]. 

### 2.4. Cardiorespiratory Fitness

Cardiorespiratory fitness (CRF) was assessed using the 20 m shuttle run protocol. The test was conducted according to Leger et al. (1988) [33] in an indoor sports gymnasium, requiring participants to run between two marks set 20 m apart. The initial velocity was defined at 8.5 km·h^−1^ and increased by 0.5 km·h^−1^ each minute. The participants were instructed to complete the shuttle and keep pace according to the audio announced on a tape player. During the test, participants were verbally encouraged, and the number of laps was retained for the analysis. The test was concluded when participants stopped the run due to fatigue or failed to reach the end marks on two consecutive occasions. 

### 2.5. Horizontal Jump

The participants were instructed to jump from the starting line as far as possible. The standing jump score was determined by the distance between the starting line and the heel. Two measures were performed, and the average value was considered in the analysis. 

### 2.6. Flexibility

The sit and reach protocol was used to estimate the flexibility of lower limbs. Participants positioned their heels on the bench and flexed the trunk four times. On the last repetition, the position needs to sustain the trunk for at least one second. Two trials were performed on the right and left sides of the body.

### 2.7. Data Analysis

Descriptive statistics (mean ± standard deviation) were calculated separately for each CA group and sex. An independent *t*-test was used to compare males and females on body size, waist circumference, %FM and fitness tests. The BMI of individual participants was plotted relative to US reference data [31]. Data regarding body composition and fitness indicators are presented as box–whisker diagrams. Fat mass percentage [34], waist circumference [35], 20 m shuttle run [36], horizontal jump [37] and flexibility [16] tests were contrasted with the age and sex-specific cut-off points adopted by FITESCOLA. In addition, the 50th percentiles (median) of the 20 m shuttle run and flexibility tests were obtained from Portuguese data [14] and plotted on the box–whisker diagram. Statistical analyses were performed with SPSS version 20.0 (SPSS Inc., IBM Company, New York, NY, USA) and Graphpad Prism (version 5.00 for Windows, GraphPad Software, San Diego, CA, USA). The alpha level was set at 0.05.

## 3. Results

The participants completed a questionnaire about participation in organized or school sports. Approximately 33% of the participants reported participating in organized sports, while 25% were involved in school sports. Descriptive statistics for body size (mean and standard deviation) by age group and sex are shown in Table 1. Males were significantly taller than females from 13 years to adulthood. Differences in fat mass percentage persisted after 14 years, with boys significantly leaner than females. At 18 years old, males are +12.7 cm taller (t = 7.750, *p* < 0.001) and +10.3 kg heavier (t = 2.707, *p* < 0.011) than females. Variation in fitness tests by age and sex, is presented in Table 2. Girls obtained better results in flexibility tests. Contrasting results were obtained in the 20 m shuttle run and horizontal jump. 

Regarding BMI, 168 male participants were classified as healthy, whereas the numbers of adolescents categorized as overweight and obese were 39 and 35, respectively (Figure 1). Corresponding data for girls showed that 201, 41, and 16 were classified as healthy, overweight, and obese, respectively (Figure 2). 

In general, median estimates of fat mass percentage in males and females were at or below the cut-off points (Figure 3A,C). The trend was similar for waist circumference among girls (Figure 3D), while systematically lower waist circumference values were found among boys compared with reference data (Figure 3C). 

Among boys, the medians of the number of laps performed in the 20 m shuttle run test were above the cut-off points from 10 to 15 years. However, in late adolescence, the median values were systematically lower than FITESCOLA cut-off points (Figure 4A). The medians of the present study were lower than the 50th percentile found in Portuguese children and adolescents. Sit and reach performance was compared with FITESCOLA and Portuguese-specific data (Figure 4C,D). Among girls, the median number of laps performed in the cardiorespiratory fitness test lagged behind the cut-off points (Figure 5A). In contrast, the horizontal long jump (Figure 5B) and flexibility (Figure 5C,D) tests were above the Portuguese population cut-off points and 50th percentiles.

## 4. Discussion

The current study investigated the variation in body composition and fitness levels of Madeira youth. The frequencies of overweight (boys: 15%; girls: 16%) and obese (boys: 14%; girls: 6%) participants in the current study was considerable. As expected, males and females varied in body size and composition as well as in fitness indicators. Differences among groups were particularly noted at 12.0 years old. The median values of the 20 m shuttle run indicated that male late adolescents and female youth did not meet the reference values provided by the FITESCOLA battery and also those who were extrapolated for Portuguese population [14]. Although inter-individual variability in the horizontal jump and flexibility was observable, the median values of these functional capacities were at or above the reference cut-off points.

The percentages of overweight and obese participants classified according to BMI were not comparable to those reported in previous research with Madeira adolescents [38] neither in the Portugal mainland [4]. A follow-up study examined the tracking of fatness after seven years of the initial measurement in males and females from the Madeira Island. At 12, 15, and 16 years old, the prevalence of males overweight was 17%, 14%, and 7%, respectively. Corresponding data for girls were 11%, 16%, and 11%, respectively. The percentages of obese males in the previous age groups ranged from 1% to 6%, whereas among females, obesity ranged from 0% to 2% [38]. In Portugal, the National Food, Nutrition and Physical Activity Survey, in 2015–2016 found prevalence of pre-obesity in about 18% of children and 24% in adolescents [4]. Differences across studies may be explained by the lower rate of sport participation reported in the participants of the current research. Among 2666 females and 4220 males aged from different regions of Portugal, prevalence of sport participation ranged from 67.4% to 71.0% and 34.4% to 46.5%, respectively [39]. Of interest, lower rates of participation in sport were noted in the present study (school sport: 25%; organized sport: 33%). The use of BMI as an indicator of overweight and obesity has recognized limitations [40], thereby waist circumference and fat mass percentage derived from the Slaughter et al. [32] equation are often used in studies to assess body composition [15,24].

Waist circumference was longitudinally associated with systolic blood pressure and the mean arterial pressure in a longitudinal study that included 1089 children and 787 adolescents [41]. The cut-off points for waist circumference were developed based on the relationship with metabolic syndrome criteria [35]. Among males, the median values were below the reference data while the waist circumference of females was closely related to the cut-off points defined by the FITESCOLA. Interestingly, the median values of fatness in the present study were systematically lower than cut-off points but a substantial inter-individual variability was noted. Indeed, a significant proportion of boys and girls were classified as unhealthy, according to %FM. Few studies have positioned the samples [14,22] relative to healthy values; however, the impact of fatness in fitness is commonly studied [24,42,43]. Fat mass, estimated by bioimpedance explained combining with age and sex, explained 41% of cardiorespiratory fitness, whereas fat mass and body mass index explained 36% of the standing long jump in a cross-sectional sample of 225 adolescents aged 12–17 years old [43]. Consistent findings were obtained in 1223 Brazilian adolescents aged 10–17 years old; participants classified as fatter (according to FITNESSGRAM battery) had a larger to be classified as unfit for cardiorespiratory fitness [42].

The issue of cardiorespiratory fitness has received particular consideration in pediatric sciences [44,45,46] given the association with success in sport, cardiometabolic, physical and mental health [12]. The median of the 20 m shuttle run within age groups was generally below the cut-off points. Contradictory findings were obtained in the Portuguese Midlands data where 59.2% of girls and 63.1% boys were, on average, classified as healthy considering the 20 m shuttle run test [14]. These results were not sparingly given the lower rates of sport participation reported in the current sample. The probability of classifying 310 participants aged 10–18 years as active was explained by the level of engagement in competitive sports [47]. In parallel, longitudinal data of 1286 participants showed that participants categorized as consistently fit substantially increased the probability of better scores in academic achievement [8]. These issues claim special attention from institutional and governmental bodies. Mainly, the promotion of participation in organized sports and the frequent monitorization of body composition and fitness levels frequently. Nevertheless, the definition of cut-off points was based on data from the National Health and Nutrition Examination Survey using a treadmill protocol [36]. Thus, taking into account the extrapolation of these standards to the FITESCOLA and the conversion to laps, the present findings should be interpreted with caution.

Comparisons among boys and girls found substantial differences in body size, composition and physical fitness. Boys were taller and heavier than girls after 13 years old. Of note, mean ages at peak height velocity (PHV) of European samples varied more among boys than girls [48]. Furthermore, boys and girls gained, on average, 7.1 to 9.1 cm·year^−1^ and 8.2 to 10.3 cm·year^−1^, respectively [48]. Differences in body mass and composition were noted after PHV, which is consistent with previous studies in youth participants [34]. Data from Saskatchewan Pediatric Bone Mineral Accrual Study based on DXA showed that weight and fat mass gains among females occurred +0.5 years and +0.8 years after PHV. In contrast, among boys, correspondent gains are observed at +0.4 years and +0.6 years after PHV. Regarding physical fitness, the present results were comparable to those obtained in Portuguese samples [14,49]. The distance completed in the 12 min run was systematically higher in boys than girls from Madeira Growth Study [49]. Portuguese male children and adolescents performed better in the 20 m shuttle run, curl-ups, and push-ups as opposed to females. In contrast, females showed better results on the seat-and-reach protocol [14]. Boys and girls of the current study were plotted at or above the reference values for flexibility protocol. An adequate level of flexibility during adolescence is especially important since it is related to the level of fitness among youth [25].

The present study has limitations that should be recognized. Samples sizes particularly at early adolescence (i.e., 10 and 11 years old) were limited. Given the cross-sectional design of this study, extrapolations of these findings need particular attention. The battery of FITESCOLA used other protocols to measure cardiovascular (1609 m running) and neuromuscular fitness (push-ups, curl-ups, vertical jump, agility, velocity, shoulder flexibility) which were not considered in the current research. Participation in organized sport was obtained by questionnaire. Obviously, future studies need to include information about training characteristics which allow to examine the impact of sport on fitness and body composition with more precise detail. Finally, a recent study [49] reported the impact of socioeconomic status on size and adiposity, which is not considered in the current research.

## 5. Conclusions

In summary, the prevalence of overweight and obesity expressed by BMI in Madeira was highest compared with Portuguese surveys, which claim attention from local organizations. The levels of unfit participants expressed by the FM equation [32] also require specific exercise and nutritional interventions among youth. In addition, promoting participation in organized sports is crucial to reducing these fat levels.

## Figures and Tables

**Figure 1 children-09-01833-f001:**
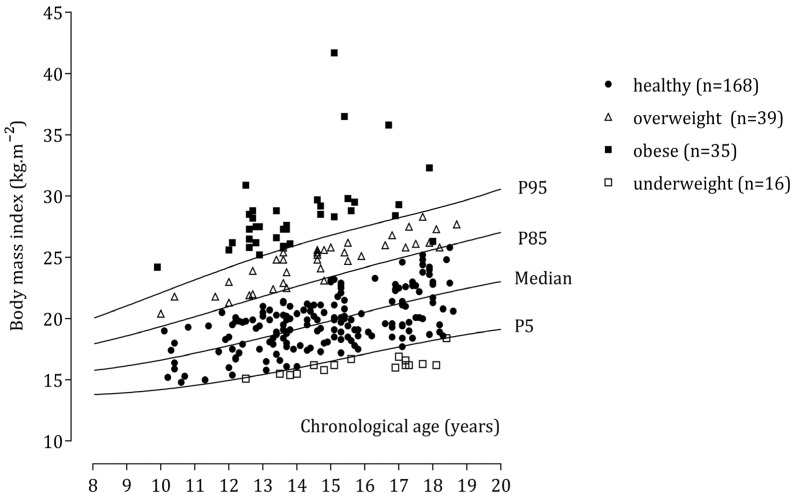
Body mass index of individual male school participants plotted relative to U.S. data.

**Figure 2 children-09-01833-f002:**
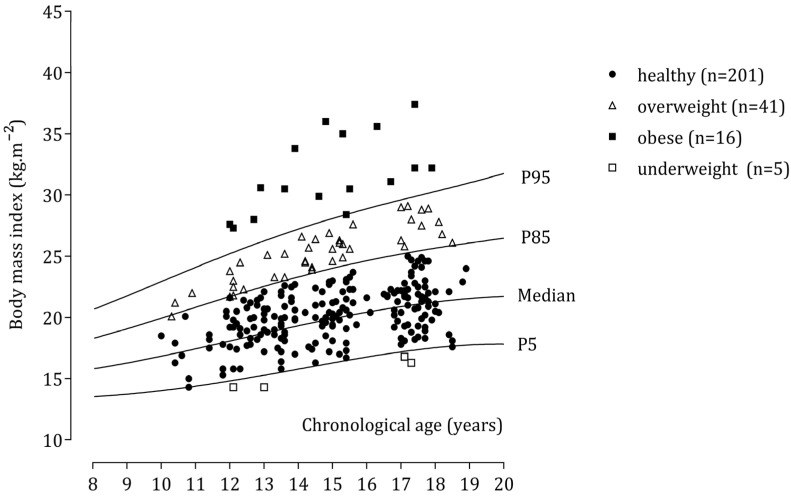
Body mass index of individual female school participants plotted relative to U.S. data.

**Figure 3 children-09-01833-f003:**
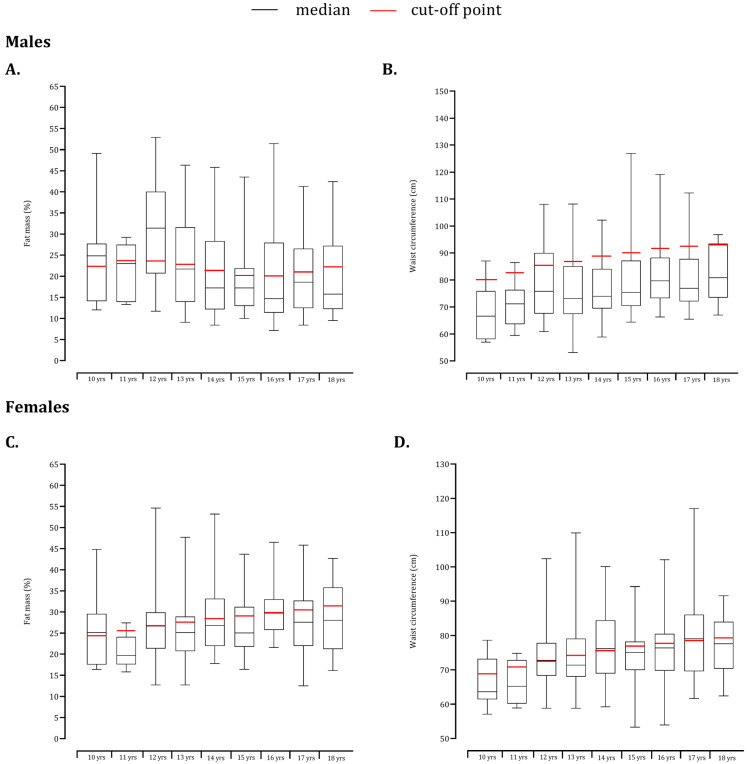
Box and whisker plots of the age-groups plotted relative to cut-off standards for fat mass (males: (**A**); females: (**C**)) and waist circumference (males: (**B**); females: (**D**)).

**Figure 4 children-09-01833-f004:**
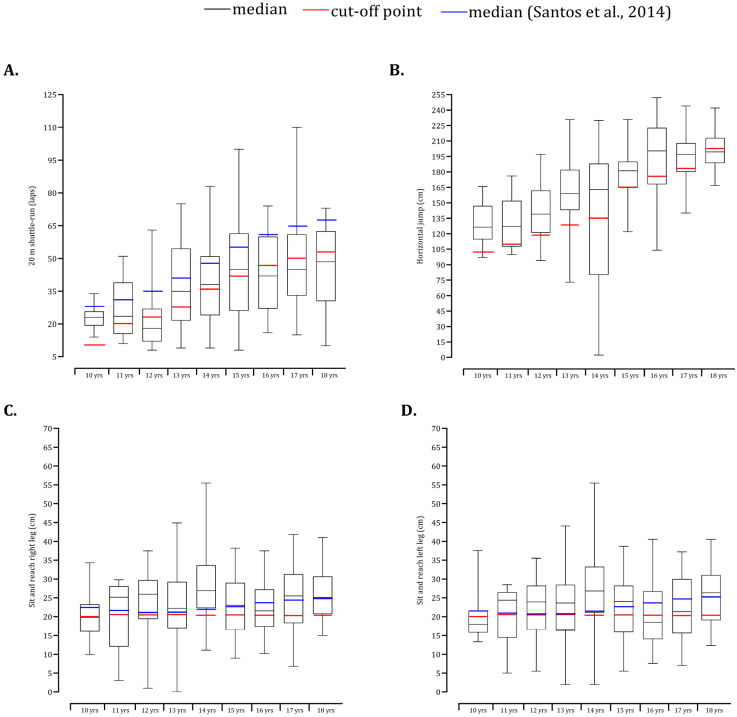
Box and whisker plots of school male participants plotted relative to cut-off standards for fitness battery (20 m shuttle-run: (**A**); horizontal jump: (**B**); sit and reach right leg: (**C**); sit and reach left leg: (**D**)) [14].

**Figure 5 children-09-01833-f005:**
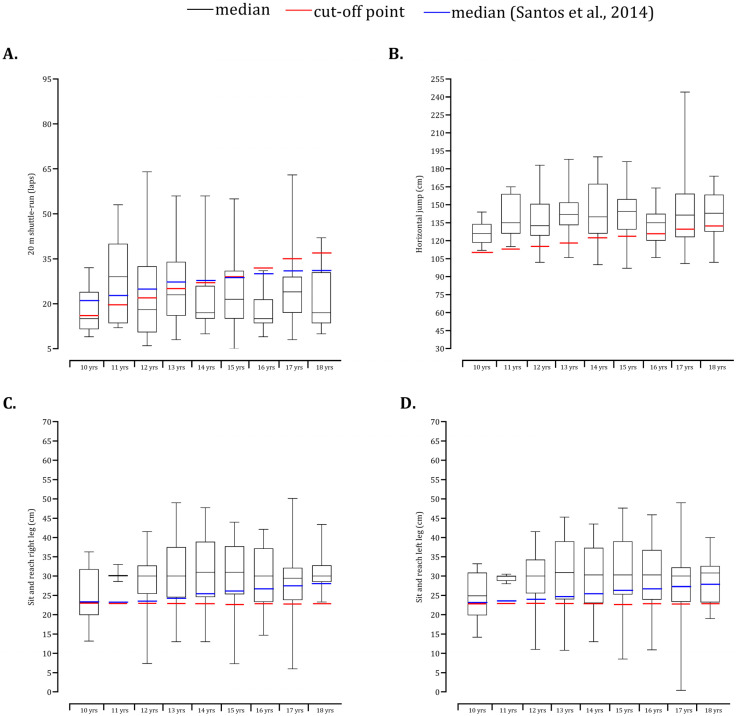
Box and whisker plots of school female participants plotted relative to cut-off standards for fitness battery (20 m shuttle-run: (**A**); horizontal jump: (**B**); sit and reach right leg: (**C**); sit and reach left leg: (**D**)) [14].

**Table 1 children-09-01833-t001:** Descriptive statistics of body size and comparisons by sex within age groups.

CA Group, Years	Variables	Boys	Girls	Mean Differences	
		Mean ± SD	Mean ± SD	t	*p*
10.0–10.9	Height, cm	142.2 ± 6.7	143.0 ± 5.6	0.308	0.761
n = 12 boys	Weight, kg	37.0 ± 8.5	37.4 ± 6.6	0.116	0.909
n = 10 girls	BMI, kg·m^−2^	18.1 ± 2.9	18.2 ± 2.6	0.072	0.943
	Waist circumference, cm	67.8 ± 9.8	66.1 ± 7.2	0.460	0.650
	Fat mass, %	24.0 ± 10.2	25.7 ± 8.9	0.423	0.677
11.0–11.9	Height, cm	147.6 ± 10.2	149.4 ± 5.0	0.464	0.650
n = 6 boys	Weight, kg	41.2 ± 9.6	39.7 ± 5.9	0.380	0.710
n = 9 girls	BMI, kg·m^−2^	18.7 ± 2.4	17.7 ± 1.9	0.893	0.388
	Waist circumference, cm	71.1 ± 9.1	66.1 ± 6.1	1.259	0.230
	Fat mass, %	21.6 ± 6.6	20.7 ± 3.9	0.309	0.762
12.0–12.9	Height, cm	155.8 ± 7.7	156.2 ± 4.9	0.271	0.787
n = 37 boys	Weight, kg	53.5 ± 12.9	50.8 ± 10.5	1.038	0.303
n = 40 girls	BMI, kg·m^−2^	21.9 ± 4.5	20.7 ± 3.4	1.408	0164
	Waist circumference, cm	79.3 ± 13.1	74.2 ± 9.3	1.980	0.052
	Fat mass, %	31.3 ± 12.4	26.4 ± 7.2	2.101	0.040
13.0–13.9	Height, cm	161.3 ± 8.4	157.4 ± 5.2	2.703	0.008
n = 49 boys	Weight, kg	54.6 ± 12.6	51.2 ± 10.5	1.408	0.163
n = 43 girls	BMI, kg·m^−2^	20.8 ± 3.6	20.6 ± 3.5	0.379	0.706
	Waist circumference, cm	76.9 ± 12.4	73.4 ± 9.5	1.498	0.138
	Fat mass, %	23.4 ± 10.3	25.7 ± 7.5	1.238	0.219
14.0–14.9	Height, cm	164.7 ± 7.3	158.5 ± 5.9	3.650	0.001
n = 34 boys	Weight, kg	57.2 ± 11.8	56.2 ± 12.8	0.321	0.749
n = 29 girls	BMI, kg·m^−2^	21.0 ± 3.9	22.2 ± 4.3	1.174	0.245
	Waist circumference, cm	77.2 ± 10.8	76.6 ± 10.3	0.241	0.810
	Fat mass, %	21.0 ± 10.3	28.9 ± 9.8	3.106	0.003
15.0–15.9	Height, cm	169.3 ± 7.1	160.0 ± 6.2	6.327	<0.001
n = 45 boys	Weight, kg	63.9 ± 16.0	57.2 ± 9.8	2.272	0.026
n = 40 girls	BMI, kg·m^−2^	22.2 ± 5.1	22.3 ± 3.8	0.154	0.878
	Waist circumference, cm	80.8 ± 14.1	74.8 ± 8.2	2.430	0.018
	Fat mass, %	19.2 ± 7.9	26.4 ± 6.5	4.596	<0.001
16.0–16.9	Height, cm	173.8 ± 6.0	161.6 ± 6.7	5.504	<0.001
n = 16 boys	Weight, kg	67.8 ± 13.8	59.0 ± 12.3	1.939	0.062
n = 17 girls	BMI, kg·m^−2^	22.5 ± 4.9	22.5 ± 4.3	0.034	0.973
	Waist circumference, cm	81.8 ± 13.5	76.3 ± 10.4	1.310	0.200
	Fat mass, %	19.8 ± 11.6	30.4 ± 6.4	3.266	0.004
17.0–17.9	Height, cm	172.9 ± 6.6	162.0 ± 6.0	8.581	<0.001
n = 41 boys	Weight, kg	66.0 ± 13.0	59.7 ± 13.1	2.369	0.020
n = 62 girls	BMI, kg·m^−2^	22.0 ± 3.9	22.6 ± 4.1	0.763	0.447
	Waist circumference, cm	80.6 ± 11.3	79.0 ± 11.2	0.722	0.472
	Fat mass, %	20.5 ± 10.0	29.3 ± 10.6	4.261	<0.001
18.0–18.9	Height, cm	174.7 ± 4.9	162.0 ± 3.9	7.750	<0.001
n = 18 boys	Weight, kg	68.4 ± 11.7	58.1 ± 8.4	2.707	0.011
n = 13 girls	BMI, kg·m^−2^	22.4 ± 3.4	22.2 ± 3.3	0.160	0.874
	Waist circumference, cm	82.4 ± 9.9	77.3 ± 8.8	1.496	0.146
	Fat mass, %	19.4 ± 1	28.1 ± 8.9	2.497	0.018

Abbreviations: BMI, body mass index; cm, centimeters; kg, kilogram; SD, standard deviation.

**Table 2 children-09-01833-t002:** Descriptive statistics of physical fitness tests and comparisons by sex within age groups.

CA Group, Years	Variables	Boys	Girls	Mean Differences	
		Mean ± SD	Mean ± SD	t	*p*
10.0–10.9	Shuttle run, laps	23 ± 6	18 ± 8	1.858	0.078
n = 12 boys	Horizontal jump, cm	129 ± 21	127 ± 10	0.394	0.699
n = 10 girls	Sit and reach RL, cm	20.1 ± 6.2	25.1 ± 7.2	1.733	0.098
	Sit and reach LL, cm	19.4 ± 6.5	25.0 ± 6.0	2.082	0.050
11.0–11.9	Shuttle run, laps	27 ± 14	29 ± 14	0.287	0.779
n = 6 boys	Horizontal jump, cm	131 ± 28	140 ± 18	0.841	0.416
n = 9 girls	Sit and reach RL, cm	20.9 ± 10.2	30.2 ± 1.2	2.222	0.076
	Sit and reach LL, cm	20.9 ± 8.6	29.4 ± 0.8	2.412	0.060
12.0–12.9	Shuttle run, laps	23 ± 14	22 ± 13	0.056	0.956
n = 37 boys	Horizontal jump, cm	143 ± 26	136 ± 19	1.164	0.249
n = 40 girls	Sit and reach RL, cm	23.8 ± 7.5	28.9 ± 5.9	3.336	0.001
	Sit and reach LL, cm	22.4 ± 7.2	29.2 ± 6.1	4.447	<0.001
13.0–13.9	Shuttle run, laps	37 ± 21	26 ± 12	3.205	0.002
n = 49 boys	Horizontal jump, cm	161 ± 33	142 ± 18	3.588	0.001
n = 43 girls	Sit and reach RL, cm	22.3 ± 9.1	30.6 ± 8.3	4.488	<0.001
	Sit and reach LL, cm	21.8 ± 8.9	30.7 ± 9.0	4.743	<0.001
14.0–14.9	Shuttle run, laps	40 ± 19	22 ± 11	4.583	<0.001
n = 34 boys	Horizontal jump, cm	173 ± 39	143 ± 23	3.604	0.001
n = 29 girls	Sit and reach RL, cm	27.9 ± 9.5	30.9 ± 9.3	1.267	0.210
	Sit and reach LL, cm	26.8 ± 10.0	30.0 ± 8.6	1.350	0.182
15.0–15.9	Shuttle run, laps	44 ± 21	24 ± 12	5.609	<0.001
n = 45 boys	Horizontal jump, cm	178 ± 27	144 ± 21	6.333	<0.001
n = 40 girls	Sit and reach RL, cm	23.2 ± 7.4	30.9 ± 8.5	4.483	<0.001
	Sit and reach LL, cm	22.2 ± 8.2	30.5 ± 9.3	4.352	<0.001
16.0–16.9	Shuttle run, laps	43 ± 19	17 ± 7	5.324	<0.001
n = 16 boys	Horizontal jump, cm	193 ± 41	134 ± 16	5.500	<0.001
n = 17 girls	Sit and reach RL, cm	22.4 ± 7.0	29.9 ± 8.0	2.846	0.008
	Sit and reach LL, cm	20.7 ± 9.2	30.2 ± 8.4	3.073	0.004
17.0–17.9	Shuttle run, laps	49 ± 22	25 ± 11	6.443	<0.001
n = 41 boys	Horizontal jump, cm	193 ± 24	145 ± 28	9.275	<0.001
n = 62 girls	Sit and reach RL, cm	25.2 ± 8.5	27.9 ± 7.2	1.754	0.082
	Sit and reach LL, cm	22.6 ± 8.6	27.6 ± 8.0	3.003	0.003
18.0–18.9	Shuttle run, laps	47 ± 19	22 ± 11	4.328	<0.001
n = 18 boys	Horizontal jump, cm	202 ± 21	142 ± 20	8.093	<0.001
n = 13 girls	Sit and reach RL, cm	25.8 ± 6.9	31.0 ± 4.7	2.369	0.025
	Sit and reach LL, cm	25.2 ± 8.0	29.0 ± 6.5	1.422	0.166

Abbreviations: cm, centimeters; LL, left leg; RL, right leg; SD, standard deviation.

## Data Availability

The data presented in this study are available upon request from the corresponding author.

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
