# Peer review of "Body Composition and Physical Fitness in Madeira Youth"

_children, 2022, doi:10.3390/children9121833_

Round 1
Reviewer 1 Report
Thank you for allowing me to review this manuscript. The study aimed to describe the physical fitness and body composition of young males and females in Madeira, as well as compare these to reference standards. The study demonstrates that the proportion of youth classified as overweight and obese are higher than estimates from other regions.
Firstly, I would like to commend the authors on data collection across a modest sample size. It is certainly no easy task. Secondly, the written English of the manuscript is mostly clear, which deserves praise as I assume they are not native English speakers. However, I have many major comments regarding the scientific aspects of the manuscript which are highlighted in detail below. I believe such surveillance data in youth from Madeira are important, though there is a lot to be improved with the depth and clarity of the manuscript. Additionally, I don’t think the manuscript is wholly appropriate for the special issue “Global Trends in Insufficient Physical Activity among Adolescents”.
General Comments
The authors reference their study sample as ‘children’ despite chronological ages ranging from 8.8 – 18.9 years. I would argue that this sample is inclusive of children and adolescents (and even young adults) and therefore consistent terminology should be adopted to reflect this.
The authors initialise “chronological age” to “CA”. I am not sure the “CA” initialism is warranted given how little it is used throughout the manuscript.
One of the purposes of the study was to ‘describe’ the measured parameters in males and females. Therefore, is it necessary to perform any statistical analyses? Do the authors believe that the male-female differences (that are consistently demonstrated across many independent studies and samples) would be any different in Madeira?
Sample size – total sample is large, for which the authors should be commended on their hard work. However, representation for some age groups is lacking quite a lot (i.e. 11 year olds) which inevitably affects the ability to make appropriate comparisons.
Introduction
Overall, the rationale for the study has not been fully developed in a clear manner. Below are specific comments and recommendations. However, there are instances where relevant, key sources have not been included in the introduction to help provide the relevant background information. The authors also do not emphasise the original aspects of the study as they progress through the introduction.
Page 1, line 47: Clarify what proportion were classified as pre-obese and obese. Were these data also stratified by children/adolescents or were they considered as one?
Page 2, line 48: Is this the overall prevalence in Madeira? This statement, whilst necessary, feels out of place. I would consider moving it after the sentence regarding the Portuguese survey.
Page 2, line 48-50: I understand the authors are trying to highlight the lack of data on Madeira youth, but there are other sources with useful information that they have not considered (e.g. https://link.springer.com/article/10.1007/s10389-007-0109-1).
Page 2, line 54-56: The authors state that research has documented a “potential effect of physical activity on obesity” but this is vague. What is that effect? Direction? Magnitude?
Page 2, lines 54-67: It is unclear if the focus of this paragraph is physical activity or physical fitness. They are two entirely different qualities. Given that the study measures physical fitness, it is surprising this is not discussed more in the introduction. There are numerous studies documenting physical fitness among global youth samples (e.g. https://pubmed.ncbi.nlm.nih.gov/27208067/) as well as the associations between fitness and health (e.g. https://bjsm.bmj.com/content/50/23/1451.abstract; https://pubmed.ncbi.nlm.nih.gov/28254744/#:~:text=Results%3A%20Ten%20studies%20reported%20criterion,%25%C2%B116%25%20among%20boys.) Such sources would warrant inclusion and discussion of an experiment like this one.
Page 2, lines 68-77: This paragraph is difficult to follow. It is not clear which two studies the authors are referring to, as citations have not been given in the relevant places. I would assume it is citation number 15 from the previous paragraph, but this is not made explicitly clear.
Page 2, line 72: “Meantime” should be replaced with “Meanwhile”.
Do the authors have a hypothesis/hypotheses? Or is the study primarily exploratory in nature? Either way, it is worth stating for clarity.
Methods
Page 3, line 118 & 119: “German” should be “Germany”.
Page 3: Was there a particular reason for body mass to be measured to the nearest 0.5 kg? Most digital scales are accurate enough to measure to 0.1 kg.
Page 3: Why were American children and adolescents chosen as the reference for BMI? There are other reference data (e.g. UK) that could have also been used. It would be interesting to see the comparisons with other reference data, to see if the patterns remain the same.
Page 3, line 126: Please specify which skinfold sites were used. I assume it was triceps and subscapular, but it should be stated clearly so that others could replicate the methods if required. How many times were these measured? Can the authors justify why a fat mass % calculation has been used? The validity of such calculations from skinfold measurements has been questioned previously (see; https://www.mdpi.com/2072-6643/13/4/1075 & https://www.ncbi.nlm.nih.gov/pmc/articles/PMC1511327/ ). Typically, the sum of skinfolds serves as a more appropriate measure with fewer sources of error.
Page 3: Were there any measurements in place to determine if a maximum effort was achieved on the 20m shuttle run? (i.e. heart rate measurement). If not, how are the authors confident that participants ran until physiological maximum?
Page 4, line 152: For the data analysis, I assume the alpha level for significance testing was set to 0.05? Worth stating just to be clear.
Page 4, lines 156-158: Please state how the empirical data was ‘compared’ to the reference data where appropriate (i.e. were statistical tests performed?).
The authors could also consider comparing the shuttle run performance to a world-wide dataset (https://pubmed.ncbi.nlm.nih.gov/27208067/)
Page 4, lines 152-160: Given that comparisons between males and females were conducted across all age groups, for many parameters, can the authors explain whether or not they did any adjustments to help with error control? If the authors don’t feel it necessary, then the nature of multiple tests should be caveated in the discussion of findings.
Results
Page 7, Figure 2: The figure legend says “male” participants but it should be “female”.
Page 8: Figure 3A: The reference standards cited for BF% only refer to children of 12 years and above. Were the 10 & 11 year olds compared to this? If so, do the authors think this is a valid comparison?
- Furthermore, the reference standards for shuttle run performance are from a paper using direct measurement of VO2max. Can the authors clarify how they compare shuttle run (number of laps) to these reference standards?
Page 8 & 9: The comparison of the study data with previous data/reference standards is largely descriptive. The authors mention that in some instances, the median of the sample is higher/lower than the reference, but there is no indication on the magnitude of this difference. It would be worth knowing: how much different are the study data compared to reference standards? Is this meaningful? (statistically but also practically). Currently, it is largely descriptive and does not provide much insight.
Discussion
Overall, the discussion lacks depth/critical analysis of the study findings. It would be useful to extrapolate on certain points that have been made in the discussion. For example, the authors mention the different body composition comparisons (BMI, FM%) but did not discuss the relevant merits of these methods and whether or not they represent what they are actually measuring.
There was no mention/discussion of the waist circumference results.
Additionally, the structure of the discussion is not logical and lacks clarity. Given that the study had 2 primary aims (comparison of boys and girls; comparison with reference standards) it would make sense to discuss the findings systematically, with reference to these study aims.
Page 11, line 233-235: I am struggling to see the relevance of this sentence within this particular paragraph. The paragraph has been discussing the measurements of body composition, but then a reference to fitness and academic achievement has been made.
Page 11, lines 236-245: This discussion of PHV seems out of place. Of course, PHV is a relevant consideration but I do not see the clear link for discussing it here. It is also unclear how any of the empirical study data relates to the statements made here.
Author Response
Reviewer 1
Thank you for allowing me to review this manuscript. The study aimed to describe the physical fitness and body composition of young males and females in Madeira, as well as compare these to reference standards. The study demonstrates that the proportion of youth classified as overweight and obese are higher than estimates from other regions.
Firstly, I would like to commend the authors on data collection across a modest sample size. It is certainly no easy task. Secondly, the written English of the manuscript is mostly clear, which deserves praise as I assume they are not native English speakers. However, I have many major comments regarding the scientific aspects of the manuscript which are highlighted in detail below. I believe such surveillance data in youth from Madeira are important, though there is a lot to be improved with the depth and clarity of the manuscript. Additionally, I don’t think the manuscript is wholly appropriate for the special issue “Global Trends in Insufficient Physical Activity among Adolescents”.
AUTHORS: Appreciate your commentaries and suggestions. According to your points, as well as reviewer 2 suggestions, the authors included them in the manuscript. The authors will write to the editorial board to retrieve the paper from the special issue and to submit for overall appreciation in Children. Thanks.
General Comments
The authors reference their study sample as ‘children’ despite chronological ages ranging from 8.8 – 18.9 years. I would argue that this sample is inclusive of children and adolescents (and even young adults) and therefore consistent terminology should be adopted to reflect this.
AUTHORS: Thanks. There was an error in the age range. This point was adjusted.
The authors initialise “chronological age” to “CA”. I am not sure the “CA” initialism is warranted given how little it is used throughout the manuscript.
AUTHORS: Thanks, adjusted.
One of the purposes of the study was to ‘describe’ the measured parameters in males and females. Therefore, is it necessary to perform any statistical analyses? Do the authors believe that the male-female differences (that are consistently demonstrated across many independent studies and samples) would be any different in Madeira?
AUTHORS: The purpose of this comparison was to discuss variations in the timing of males and females related to body size and body composition.
Sample size – total sample is large, for which the authors should be commended on their hard work. However, representation for some age groups is lacking quite a lot (i.e. 11 year olds) which inevitably affects the ability to make appropriate comparisons.
AUTHORS: Thanks, this part is clearly a main limitation, and it is now considered in the discussion.
Introduction
Overall, the rationale for the study has not been fully developed in a clear manner. Below are specific comments and recommendations. However, there are instances where relevant, key sources have not been included in the introduction to help provide the relevant background information. The authors also do not emphasise the original aspects of the study as they progress through the introduction.
AUTHORS: Authors agree with the commentary.
Page 1, line 47: Clarify what proportion were classified as pre-obese and obese. Were these data also stratified by children/adolescents or were they considered as
AUTHORS: Thanks, this point was now clarified.
Page 2, line 48: Is this the overall prevalence in Madeira? This statement, whilst necessary, feels out of place. I would consider moving it after the sentence regarding the Portuguese survey.
AUTHORS: This sentence is now clarified and organized.
Page 2, line 48-50: I understand the authors are trying to highlight the lack of data on Madeira youth, but there are other sources with useful information that they have not considered (e.g. https://link.springer.com/article/10.1007/s10389-007-0109-1).
AUTHORS: The reference to the Journal of Public Health was added to the first paragraph of the introduction as well as two references.
Ekelund, U., Sardinha, L. B., Anderssen, S. A., Harro, M., Franks, P. W., Brage, S., Cooper, A. R., Andersen, L. B., Riddoch, C., & Froberg, K. (2004). Associations between objectively assessed physical activity and indicators of body fatness in 9- to 10-y-old European children: a population-based study from 4 distinct regions in Europe (the European Youth Heart Study). The American Journal of Clinical Nutrition, 80(3), 584–590. https://doi.org/10.1093/ajcn/80.3.584
Sousa B, Oliveira B, Almeida MDV (2006) Assessment of nutritional status in 6- to 10-years-old children of the Autonomous Region of Madeira, Portugal. Public Health Nutr 9(7A):109
Page 2, line 54-56: The authors state that research has documented a “potential effect of physical activity on obesity” but this is vague. What is that effect? Direction? Magnitude?
AUTHORS: This sentence was deleted and the introduction was adjusted according to the reviewer's suggestions.
Page 2, lines 54-67: It is unclear if the focus of this paragraph is physical activity or physical fitness. They are two entirely different qualities. Given that the study measures physical fitness, it is surprising this is not discussed more in the introduction. There are numerous studies documenting physical fitness among global youth samples (e.g. https://pubmed.ncbi.nlm.nih.gov/27208067/) as well as the associations between fitness and health (e.g. https://bjsm.bmj.com/content/50/23/1451.abstract; https://pubmed.ncbi.nlm.nih.gov/28254744/#:~:text=Results%3A%20Ten%20studies%20reported%20criterion,%25%C2%B116%25%20among%20boys.) Such sources would warrant inclusion and discussion of an experiment like this one.
AUTHORS: The introduction was reformulated according to the reviewers’ feedback and new references were added. The authors believe that the introduction has considerably improved its quality.
Page 2, lines 68-77: This paragraph is difficult to follow. It is not clear which two studies the authors are referring to, as citations have not been given in the relevant places. I would assume it is citation number 15 from the previous paragraph, but this is not made explicitly clear.
AUTHORS: The introduction was rewritten according to your suggestions.
Page 2, line 72: “Meantime” should be replaced with “Meanwhile”.
AUTHORS: This part was deleted.
Do the authors have a hypothesis/hypotheses? Or is the study primarily exploratory in nature? Either way, it is worth stating for clarity.
AUTHORS: The study is descriptive and the aim was reformulated.
Methods
Page 3, line 118 & 119: “German” should be “Germany”.
AUTHORS: Thanks, adjusted.
Page 3: Was there a particular reason for body mass to be measured to the nearest 0.5 kg? Most digital scales are accurate enough to measure to 0.1 kg.
AUTHORS: It was identified an error on the report and it is now clear.
Page 3: Why were American children and adolescents chosen as the reference for BMI? There are other reference data (e.g. UK) that could have also been used. It would be interesting to see the comparisons with other reference data, to see if the patterns remain the same.
AUTHORS: Note the BMI cut-off values of the CDC are age-specific. Although the authors hypothesized to use the cut-off values of IOBT, a parallel manuscript regarding the percentiles of Madeira youth and comparisons with other different sources of data is being prepared.
Page 3, line 126: Please specify which skinfold sites were used. I assume it was triceps and subscapular, but it should be stated clearly so that others could replicate the methods if required. How many times were these measured? Can the authors justify why a fat mass % calculation has been used? The validity of such calculations from skinfold measurements has been questioned previously (see; https://www.mdpi.com/2072-6643/13/4/1075 & https://www.ncbi.nlm.nih.gov/pmc/articles/PMC1511327/ ). Typically, the sum of skinfolds serves as a more appropriate measure with fewer sources of error.
AUTHORS: The methods regarding body composition assessment were adjusted. Note, the equation is adopted by FITNESSGRAM and FITESCOLA batteries to assess body composition. Although the sum of skinfolds is useful, there are no cut-off points for this parameter considering the school batteries. The protocol was applied previously in relevant data (Coelho e Silva et al., 2014; Henriques-Neto et al., 2020; Martinho et al., 2021). Some of your questions and comments were added to the section on anthropometry.
Page 3: Were there any measurements in place to determine if a maximum effort was achieved on the 20m shuttle run? (i.e. heart rate measurement). If not, how are the authors confident that participants ran until physiological maximum?
AUTHORS: Unfortunately, this is a main limitation of the protocol. However, data quality about the 20 m shuttle is well documented (Tomkison et al., 2008).
Tomkinson GR, Olds TS. Field tests of fitness. In: Armstrong N, van Mechelen W, eds. Paediatric exercise science and medicine. New York, NY: Oxford University Press, 2008:109–28.
Page 4, line 152: For the data analysis, I assume the alpha level for significance testing was set to 0.05? Worth stating just to be clear.
AUTHORS: Thanks, adjusted.
Page 4, lines 156-158: Please state how the empirical data was ‘compared’ to the reference data where appropriate (i.e. were statistical tests performed?).
AUTHORS: Thanks, authors agree with your commentary. The term compared implies a statistical test to compare groups. Meantime, the term compared was changed by plotted and contrasted. Hope makes sense.
The authors could also consider comparing the shuttle run performance to a world-wide dataset (https://pubmed.ncbi.nlm.nih.gov/27208067/)
AUTHORS: Authors understand the preference for the paper of the British Journal of Sports Medicine. However, the current sample is not large and this point was recognized in the limitation section. In addition, the shuttle run performance is influenced by other factors (e.g. time and socio-economic status). As previously mentioned, a candidate manuscript is in preparation about the percentiles of Madeira youth data and potential sources of comparison are under discussion. Considering the preceding, the authors use the FITESCOLA or Portuguese reference data.
Page 4, lines 152-160: Given that comparisons between males and females were conducted across all age groups, for many parameters, can the authors explain whether or not they did any adjustments to help with error control? If the authors don’t feel it necessary, then the nature of multiple tests should be caveated in the discussion of findings.
AUTHORS: The comparisons between males and females are now evident in the discussion section.
Results
Page 7, Figure 2: The figure legend says “male” participants but it should be “female”.
Thanks, it is now adjusted.
Page 8: Figure 3A: The reference standards cited for BF% only refer to children of 12 years and above. Were the 10 & 11 year olds compared to this? If so, do the authors think this is a valid comparison?
- Furthermore, the reference standards for shuttle run performance are from a paper using direct measurement of VO2max. Can the authors clarify how they compare shuttle run (number of laps) to these reference standards?
Thanks for the pertinent questions. Regarding the fat mass percentage if the reviewer consulted the original reference can find in the last paragraph:
“The current study used adolescents (aged 12–19 years) to derive %BF standards. However, these values could be extrapolated to younger children using the previously created %BF centiles,9 which potentially allows for earlier identifıcation and intervention of at-risk youth if tracking of current %BF was maintained”. The reference 9 (see below) has percentiles for lower ages and was added to the analysis section. Thereby, the reference values of FITESCOLA ranged from 9-18 years.
Laurson, K. R., Eisenmann, J. C., & Welk, G. J. (2011). Body fat percentile curves for U.S. children and adolescents. American journal of preventive medicine, 41(4 Suppl 2), S87–S92. https://doi.org/10.1016/j.amepre.2011.06.044
The second question also deserves consideration. Note, the last sentence of the original paper published in 2011 stated that:
“These values could be useful in school and sport programs or clinical settings. The new FITNESSGRAM standards were based on the values developed in this paper.27”
Although the FITNESSGRAM battery provides the cut-off values as an estimation of maximal oxygen uptake (p. 6-10 and p. 6-11), the authors of FITESCOLA opted to present the cut-off values as in the previous FITNESSGRAM® standards for aerobic capacity, with the laps equivalent to the aerobic capacity. This point is not surprising given the limitations of equations to predict maximal oxygen uptake, those have received discussion in pediatric sciences (Armstrong & Welsman, 2019; Armstrong & Welsman, 2021). The authors have an original copy of FITESCOLA standards and could provide it to the reviewers.
Armstrong, N., & Welsman, J. (2021). Comment on 'Developing a New Curvilinear Allometric Model to Improve the Fit and Validity of the 20-m Shuttle Run Test as a Predictor of Cardiorespiratory Fitness in Adults and Youth'. Sports medicine (Auckland, N.Z.), 51(7), 1591–1593. https://doi.org/10.1007/s40279-021-01462-5
Armstrong, N., & Welsman, J. (2019). Clarity and Confusion in the Development of Youth Aerobic Fitness. Frontiers in physiology, 10, 979.
Page 8 & 9: The comparison of the study data with previous data/reference standards is largely descriptive. The authors mention that in some instances, the median of the sample is higher/lower than the reference, but there is no indication on the magnitude of this difference. It would be worth knowing: how much different are the study data compared to reference standards? Is this meaningful? (statistically but also practically). Currently, it is largely descriptive and does not provide much insight.
AUTHORS: The authors did not use statistical inferences because used the median (which is present on the box and whiskers plot) and comparisons were made with reference data. The paper of Santos et al. (2014) presented the comparisons when contrasting males and females. The paper is descriptive of a specific region in Portugal – where descriptive studies of physical fitness are lacking. A paragraph about Madeira's literature in youth is included in the discussion. The results were adjusted and are now more intentional.
Discussion
Overall, the discussion lacks depth/critical analysis of the study findings. It would be useful to extrapolate on certain points that have been made in the discussion. For example, the authors mention the different body composition comparisons (BMI, FM%) but did not discuss the relevant merits of these methods and whether or not they represent what they are actually measuring.
AUTHORS: The discussion was substantially improved following the organization: main findings, BMI, fatness, cardiovascular topic, males and females, limitations, and conclusion.
There was no mention/discussion of the waist circumference results.
AUTHORS: This is now included in the discussion.
Additionally, the structure of the discussion is not logical and lacks clarity. Given that the study had 2 primary aims (comparison of boys and girls; comparison with reference standards) it would make sense to discuss the findings systematically, with reference to these study aims.
Page 11, line 233-235: I am struggling to see the relevance of this sentence within this particular paragraph. The paragraph has been discussing the measurements of body composition, but then a reference to fitness and academic achievement has been made.
Page 11, lines 236-245: This discussion of PHV seems out of place. Of course, PHV is a relevant consideration but I do not see the clear link for discussing it here. It is also unclear how any of the empirical study data relates to the statements made here.
AUTHORS: Thanks, all the comments were analysed and considered by the authors.
Reviewer 2 Report
The study is interesting as it describes the sample of a specific region. However, the manuscript is very messy. There is no clear line to follow. It continually mixes up the variables and it is difficult to follow both the introduction and the discussion. The authors are requested to organise the ideas according to their purpose and to update the references, as there are hardly any recent studies included. The results are not compared with similar studies in other countries.
In the summary, the conclusions of the study should be in relation to the results obtained by the study, not the applications that these results should lead to.
In general terms, the introduction is very disorganised, it talks about body composition, then physical condition, and then mixes body composition data again.
Between lines 46 and 50, several studies are mentioned but the results of these studies are not presented and the situation to which they refer is not clarified. The wording is confusing.
Lines 68 to 79, the parrao is not understood, different studies are mentioned but it is not made clear what each one says and to which one it refers, citations are missing (line 72).
It is recommended that the authors follow a common thread for the same based on their objective.
Inclusion and exclusion criteria are not included, although excluded participants are mentioned. Please clarify this.
Line 154. Players?
The description of the statistical analysis is vague. With what software were the analyses done, what type of variables were explored? With which tests? Where was the level of significance established? Please detail.
Figure 2 also says "male". Change.
As with the introduction, the discussion is very messy.
The results of the variables are mixed up in the same paragraph. It is recommended that the authors first discuss the variables related to anthropometry and body composition and then those related to fitness. For them, I recommend a number of current studies on the subject that would be interesting to review:
For example:
Xu Y, Mei M, Wang H, Yan Q, He G. Association between Weight Status and Physical Fitness in Chinese Mainland Children and Adolescents: A Cross-Sectional Study. International Journal of Environmental Research and Public Health. 2020;17(7):2468.
Mendoza-Muñoz, M., Adsuar, J. C., Pérez-Gómez, J., Muñoz-Bermejo, L., Garcia-Gordillo, M. Á., & Carlos-Vivas, J. (2020). Influence of body composition on physical fitness in adolescents. Medicina, 56(7), 328.
Tapia-Serrano MA, Jorge M-L, David S-O, Mikel V-S, Sánchez-Miguel PA. Mediating effect of fitness and fatness on the association between lifestyle and body dissatisfaction in Spanish youth. Physiology & Behavior. 2021;232:113340.
Iglesias-Soler E, Rúa-Alonso M, Rial-Vázquez J, Lete-Lasa JR, Clavel I, Giráldez-García MA, et al. Percentiles and principal component analysis of physical fitness from a big sample of children and adolescents aged 6-18 years: the DAFIS project. Frontiers in Psychology. 2021;12:172.
Milanese C, Sandri M, Cavedon V, Zancanaro C. The role of age, sex, anthropometry, and body composition as determinants of physical fitness in nonobese children aged 6-12. PeerJ. 2020;8:e8657.
The references he uses are very old, there are much more current studies with which he compares the results.
Minor comments:
Line 32, delete a "w".
Author Response
Reviewer 2
The study is interesting as it describes the sample of a specific region. However, the manuscript is very messy. There is no clear line to follow. It continually mixes up the variables and it is difficult to follow both the introduction and the discussion. The authors are requested to organise the ideas according to their purpose and to update the references, as there are hardly any recent studies included. The results are not compared with similar studies in other countries.
AUTHORS: Appreciate your constructive message. The reference that reviewer 1 and reviewer 2 mentioned were partially included in the introduction and discussion.
In the summary, the conclusions of the study should be in relation to the results obtained by the study, not the applications that these results should lead to.
AUTHORS: Thanks, adjusted.
In general terms, the introduction is very disorganised, it talks about body composition, then physical condition, and then mixes body composition data again.
Between lines 46 and 50, several studies are mentioned but the results of these studies are not presented and the situation to which they refer is not clarified. The wording is confusing.
Lines 68 to 79, the parrao is not understood, different studies are mentioned but it is not made clear what each one says and to which one it refers, citations are missing (line 72).
It is recommended that the authors follow a common thread for the same based on their objective.
AUTHORS: The introduction was substantially adjusted following the reviewers’ feedback and it is believed that it has improved its overall quality.
Inclusion and exclusion criteria are not included, although excluded participants are mentioned. Please clarify this.
AUTHORS: The exclusion criteria refer to participants that do not complete the physical tests or the questionnaire.
Line 154. Players?
AUTHORS: Thanks, adjusted.
The description of the statistical analysis is vague. With what software were the analyses done, what type of variables were explored? With which tests? Where was the level of significance established? Please detail.
Figure 2 also says "male". Change.
AUTHORS: Thanks, adjusted.
As with the introduction, the discussion is very messy.
The results of the variables are mixed up in the same paragraph. It is recommended that the authors first discuss the variables related to anthropometry and body composition and then those related to fitness. For them, I recommend a number of current studies on the subject that would be interesting to review:
AUTHORS: The authors agree with your commentary and now organized the discussion according to the following points: main findings, BMI, fatness, cardiovascular topic, males and females, limitations, and conclusion. Hope makes sense.
For example:
Xu Y, Mei M, Wang H, Yan Q, He G. Association between Weight Status and Physical Fitness in Chinese Mainland Children and Adolescents: A Cross-Sectional Study. International Journal of Environmental Research and Public Health. 2020;17(7):2468.
Mendoza-Muñoz, M., Adsuar, J. C., Pérez-Gómez, J., Muñoz-Bermejo, L., Garcia-Gordillo, M. Á., & Carlos-Vivas, J. (2020). Influence of body composition on physical fitness in adolescents. Medicina, 56(7), 328.
Tapia-Serrano MA, Jorge M-L, David S-O, Mikel V-S, Sánchez-Miguel PA. Mediating effect of fitness and fatness on the association between lifestyle and body dissatisfaction in Spanish youth. Physiology & Behavior. 2021;232:113340.
Iglesias-Soler E, Rúa-Alonso M, Rial-Vázquez J, Lete-Lasa JR, Clavel I, Giráldez-García MA, et al. Frontiers in Psychology. 2021;12:172.
Milanese C, Sandri M, Cavedon V, Zancanaro C. The role of age, sex, anthropometry, and body composition as determinants of physical fitness in nonobese children aged 6-12. PeerJ. 2020;8:e8657.
AUTHORS: The first two references were introduced in the discussion as well as the references indicated by reviewer 1.
The references he uses are very old, there are much more current studies with which he compares the results.
AUTHORS: New references were added to the introduction and discussion.
Minor comments:
Line 32, delete a "w".
AUTHORS: This part was adjusted. Thanks.
Round 2
Reviewer 2 Report
Congratulations to the authors. The masnuscript has improved considerably, however, the conclusions section is not included, as specified in the template of the issue.
Author Response
REVIEWER: Congratulations to the authors. The manuscript has improved considerably, however, the conclusions section is not included, as specified in the template of the issue.
AUTHORS: Thanks, the section conclusion was added to the manuscript.